# IFNλ is a potent anti-influenza therapeutic without the inflammatory side effects of IFNα treatment

Sophia Davidson[1,†], Teresa M McCabe[1,†], Stefania Crotta[1], Hans Henrik Gad[2], Edith M Hessel[3], Soren Beinke[3], Rune Hartmann[2] & Andreas Wack[1,*]

## Abstract

Influenza A virus (IAV)-induced severe disease is characterized by infected lung epithelia, robust inflammatory responses and acute lung injury. Since type I interferon (IFNαβ) and type III interferon (IFNλ) are potent antiviral cytokines with immunomodulatory potential, we assessed their efficacy as IAV treatments. IFNλ treatment of IAV-infected Mx1-positive mice lowered viral load and protected from disease. IFNα treatment also restricted IAV replication but exacerbated disease. IFNα treatment increased pulmonary proinflammatory cytokine secretion, innate cell recruitment and epithelial cell death, unlike IFNλ-treatment. IFNλ lacked the direct stimulatory activity of IFNα on immune cells. In epithelia, both IFNs induced antiviral genes but no inflammatory cytokines. Similarly, human airway epithelia responded to both IFNα and IFNλ by induction of antiviral genes but not of cytokines, while hPBMCs responded only to IFNα. The restriction of both IFNλ responsiveness and productive IAV replication to pulmonary epithelia allows IFNλ to limit IAV spread through antiviral gene induction in relevant cells without overstimulating the immune system and driving immunopathology. We propose IFNλ as a non-inflammatory and hence superior treatment option for human IAV infection.

**Keywords** immunopathology; infection; influenza; interferon alpha; interferon lambda

**Subject Categories** Immunology; Microbiology, Virology & Host Pathogen Interaction

## Introduction

Influenza A virus (IAV) causes three to five million cases of severe illness and up to 500,000 deaths annually worldwide (Krammer *et al*, 2015). IAV is also capable of causing devastating pandemics, as typified by the 1918 "Spanish Flu" outbreak that

resulted in an estimated 50 million deaths (Kobasa *et al*, 2007). IAV primarily replicates in airway epithelial cells (AECs), and in mild forms of the disease, replication is restricted to the upper respiratory tract (Suzuki *et al*, 2000). Severe disease caused by IAV infection is characterized by lower respiratory tract infection, hypercytokinemia, bronchopneumonia and tissue destruction (Peiris *et al*, 2004; de Jong *et al*, 2006). How type I and III interferons (IFNs) contribute to IAV-induced pathology remains controversial (Kobasa *et al*, 2007; Baskin *et al*, 2009; Trinchieri, 2010; Durbin *et al*, 2013).

Immunization with inactivated or live vaccines matching circulating IAV strains is the best prophylactic option for protection from IAV. Yet current vaccines rarely induce broadly neutralizing antibodies and therefore are not able to induce cross-protection against heterologous IAV strains or subtypes (Krammer *et al*, 2015). Current influenza antivirals include ion channel blockers and neuraminidase inhibitors, which act directly on viral proteins (Jefferson *et al*, 2006). However, targeting IAV directly drives the emergence of drug-resistant strains due to the high natural mutation rate of IAV. Indeed, since the early 2000s, global spread of adamantane-resistant A (H3N2), oseltamivir-resistant seasonal A (H1N1) viruses and adamantane-resistant pandemic A (H1N1) viruses has been recorded (Hayden & de Jong, 2011). Worryingly, mutations in the IAV genome which confer drug resistance do not require selective pressure to be shared between IAV strains. In order to protect the population from new IAV strains and avoid development of drug resistance, regimens for activation of multiple antiviral host factors would provide improved treatments. The use of immunostimulatory biologics like IFNs would greatly help manage drug resistance, but can be limited by adverse side effects (Muir *et al*, 2014), which may be avoided by treatments acting specifically on epithelial cells, the primary cell target of influenza virus.

Type I IFNs are well-established antiviral cytokines comprised of 13 distinct IFNα genes in humans, one IFNβ gene and several other family members. All type I IFN subtypes act through a common, ubiquitously expressed, heterodimeric receptor (IFNαβR) to induce the transcription of a diverse set of genes known as IFN-stimulated genes (ISGs) (Randall & Goodbourn, 2008). In particular, type I IFN induces the expression of IFN-inducible transmembrane protein 3

1 Immunoregulation Laboratory, Mill Hill Laboratory, Francis Crick Institute, London, UK
2 Department of Molecular Biology and Genetics, Aarhus University, Aarhus, Denmark
3 Refractory Respiratory Inflammation Discovery Performance Unit, Respiratory Therapy Area, GSK, Stevenage, UK
*Corresponding author. Tel: +44 208 8162209; E-mail: andreas.wack@crick.ac.uk
†These authors contributed equally to this work

(IFITM3), IFN-regulatory factor 7 (IRF7) and orthomyxovirus resistance gene (Mx) family proteins, all *bona fide* restriction factors of IAV (Everitt *et al*, 2012; Ciancanelli *et al*, 2015; Haller *et al*, 2015).

More recently discovered, type III IFNs (IFNλ1, 2, 3 and 4) are also induced during viral infection and utilize the same JAK/STAT signalling pathway to activate an identical set of ISGs as type I IFNs (Kotenko *et al*, 2003; Sheppard *et al*, 2003). IFNλs bind to their own independent receptor complex consisting of IL-10R2 and IFNλR1. Unlike the ubiquitously expressed IFNαβR, IFNλR1 expression is largely restricted to mucosal surfaces such as the lung and gut epithelial layer (Kotenko *et al*, 2003; Pott *et al*, 2011). We recently demonstrated that both type I and III IFNs can induce an antiviral state in airway epithelia (Crotta *et al*, 2013), the principal target cell type of IAV. However, the effects of type I and III IFNs on immune cells vary markedly, resulting in qualitatively and quantitatively different inflammatory responses (Wack *et al*, 2015).

IAV possesses an exceptional ability to escape host adaptive immunity through antigenic shift and drift, making manufacture of a broadly neutralizing IAV vaccine highly difficult. Development of a treatment that will stimulate protective aspects of the host innate immune response to IAV is therefore highly desirable. In this context, type I IFN has been periodically discussed as a possible treatment for IAV during infection (Finter *et al*, 1991; McKinlay, 2001; Wang *et al*, 2014). Induction of antiviral genes by IFNs has been extensively demonstrated to restrict IAV replication *in vitro*, but a concern is the collateral induction of pathology. Type III IFN has been in clinical trials for hepatitis C and demonstrated less adverse effect compared to type I IFN (Muir *et al*, 2014) but was never tested against IAV. We therefore assessed the effects of treatment with exogenous type I and III IFNs during the course of IAV infection *in vivo* in congenic B6.A2G-Mx1 mice (Staeheli *et al*, 1985). These mice express functional Mx1, an IFN-induced protein that is central to IAV resistance in both its human and murine forms (Tumpey *et al*, 2007; Haller *et al*, 2015) but is lacking in most inbred mouse strains. In this improved influenza infection model, we demonstrate that type III IFN is the treatment of choice, as it avoids the potential inflammatory complications associated with type I IFN.

# Results

### Only type III IFN is protective when used therapeutically against IAV

IFN-mediated anti-IAV protection is achieved through the induction of antiviral ISGs in lung epithelia. To ascertain a comparable dosing of mouse type I and type III IFNs for *in vivo* treatments, we assessed ISG induction in AEC cultures. Cultures were treated for 4 h with either IFNα or IFNλ at stated concentrations and analysed for induction of traditional antiviral ISGs: Rsad2, Oasl2, Ifi203 and the potent anti-IAV gene Mx1 (Fig 1A). As expected, airway epithelia responded to both IFN types by ISG induction. We used the dose–response curves resulting from IFN titrations to determine a conversion factor for equipotency between IFNα and IFNλ regarding ISG induction (Fig EV1). Using this conversion factor, we were able to treat mice expressing functional Mx1 with equivalent doses of IFNα and IFNλ throughout our study.

Intranasal IFN treatment of mice prior to infection with IAV (strain A/Puerto Rico/8/34 H1N1) (PR8) was performed (Fig 1B). Both IFNα and IFNλ treatment ablated PR8-induced morbidity and mortality. This protection correlated with undetectable viral loads at 4 days post-infection (dpi) in these groups (Fig 1C), indicating that IFNα and IFNλ were equally able to induce an antiviral programme in the lung that blocked PR8 infectivity. Our data confirm previous studies that found that pretreatment of mice with exogenous type I IFNs (Tumpey *et al*, 2007; Cilloniz *et al*, 2012) or type III IFNs (Mordstein *et al*, 2008) inhibited replication of a range of IAV strains.

Pretreatment with IFNs is not a realistic clinical option to protect the population during a large-scale pandemic as patients seek medical intervention only after clinical onset. We therefore assessed the effectiveness of IFNα and IFNλ at ameliorating disease when administered after IAV infection (Fig 1D). Mice were infected with PR8 and treated after onset of clinical signs intranasally with IFNα, IFNλ or vehicle control on days 2, 4 and 5 post-infection. IFNλ-treated mice exhibited significantly lower mortality compared to vehicle control group. In striking contrast, IFNα treatment exacerbated disease causing higher mortality than in infected, vehicle-treated mice (Fig 1D). The divergent disease outcome between the IFN treatment groups was not due to different antiviral activity, as treatment-induced reduction in viral load in the lung was similar (Fig 1E).

### IFNλ treatment does not have the proinflammatory side effects found in IFNα treatment of IAV infection

Given that control of viral replication was comparable between IFNα and IFNλ treatments, we next sought to understand why IFNλ treatment was protective, whereas IFNα treatment was detrimental. Severe IAV-induced disease in humans is characterized by "cytokine storm", innate cell recruitment and epithelial cell damage (Peiris *et al*, 2004; de Jong *et al*, 2006). We therefore assessed whether or not IFNα treatment alters the immune response to IAV. We measured inflammatory cytokines in the bronchoalveolar lavage (BAL) fluid throughout IAV infection and found that IFNα treatment significantly increased secretion of IL-6, IP-10, MCP-1 and other proinflammatory cytokines from day 4 post-infection onwards, while the cytokine response in the IFNλ treatment group was comparable to infected, vehicle-treated control mice (Fig 2A). Furthermore, treatment with IFNα but not with IFNλ led to recruitment of higher numbers of plasmacytoid dendritic cells (pDCs) and inflammatory monocytes into infected lungs from 4 dpi (Fig 2B). The recruitment into infected lungs of other cell types such as B, NK and T cells is not changed by IFN treatment (Fig EV2A); however, increased activation markers are found on recruited lymphocytes only upon IFNα treatment (Fig EV2B). Significantly, IFNα treatment increased airway epithelial cell apoptosis, as assessed by TUNEL staining of lung sections, compared to control-treated mice, while IFNλ lowered the frequency of AEC apoptosis compared to control (Fig 2C and D). Thus, in the context of a functional immune response, further stimulation by IFNα leads to immunopathology which is not offset by IFNα-enhanced antiviral ISG induction increasing protection. In contrast, IFNλ treatment does not augment proinflammatory cytokine secretion during IAV infection and, importantly, results in a significant decrease in IAV-induced AEC apoptosis.

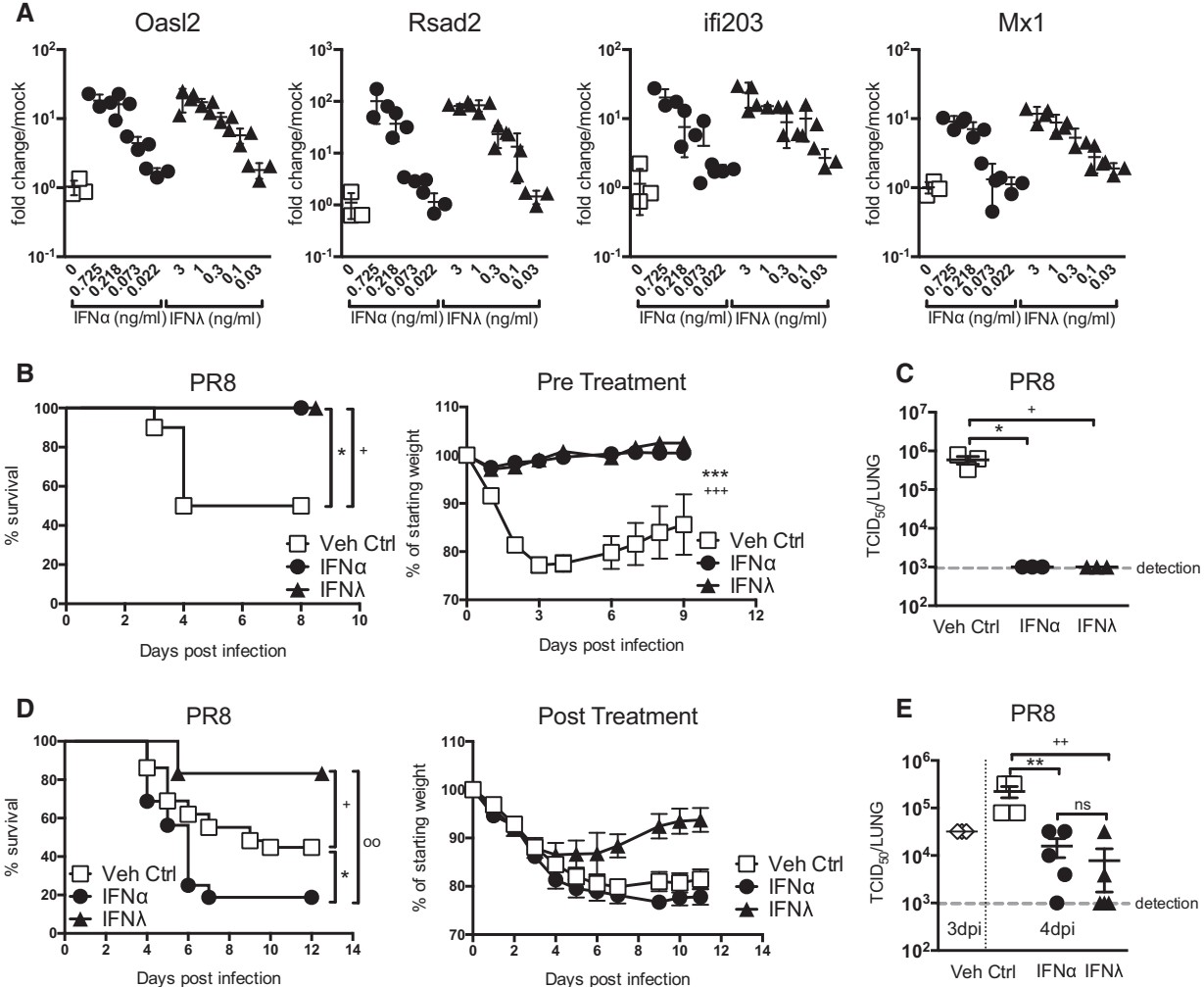

**Figure 1. Therapeutic administration of IFNα and IFNλ differentially influences the outcome of IAV-induced disease.**

A   Relative antiviral activity of IFNα (circles) or IFNλ (triangles). AEC cultures were stimulated for 4 h with stated IFN at specified concentrations (ng/ml) and induction of indicated ISGs was assessed by qPCR (data shown are representative of four independent experiments, *n* = 3–4).

B, C   Mice were pretreated with equivalent doses of IFNα (1.45 μg/50 μl) or IFNλ (2.6 μg/50 μl) or Veh Ctrl (squares, 50 μl PBS) 24 h prior to infection with PR8; weight loss and survival were assessed throughout infection (B), and viral load (C) assessed at 4 dpi (data shown are representative of two independent experiments, *n* = 8–10 (B), *n* = 3 (C)).

D, E   Mice were infected with PR8 and treated with equivalent doses of IFNα or IFNλ or Veh Ctrl at days 2, 4 and 5 post-infection; survival and weight loss were monitored (D, data pooled from 4 independent experiments, *n* = 12–29) and viral load assessed at 4 dpi (E) (data representative of two independent experiments, *n* = 3–5).

Data information: Significance assessed by log-rank (Mantel–Cox) test (survival), two-way ANOVA (weight loss) and unpaired *t*-tests (viral load). *indicates IFNα:Veh Ctrl, [+]indicates IFNλ:Veh Ctrl, and °indicates IFNα:IFNλ. *$P$ = 0.0236, [+]$P$ = 0.0236 ***$P$ < 0.0001, [+++]$P$ < 0.001 (B); *$P$ = 0.012, [+]$P$ = 0.012 (C); *$P$ = 0.0443, [+]$P$ = 0.035, [°°]$P$ = 0.0015 (D); **$P$ = 0.0081, [++]$P$ = 0.0066 (E). Symbols on the right of graphs indicate significance of whole curve. Graphs show mean ± SEM.

## IFNα but not IFNλ has proinflammatory effects on immune cells

To understand why treatments with IFNα or IFNλ during IAV infection differed so much in the ability to protect mice from IAV-induced morbidity and mortality, we assessed the response of specific cell types present in the lung during IAV infection to IFN stimulation. We first evaluated the response to IFN treatment in AEC cultures, as they are the primary infection targets of IAV. While IFNα and IFNλ treatment induced ISGs in AEC cultures (Fig 1A), neither IFNα nor IFNλ stimulation provoked secretion of inflammatory cytokines by these cells (Fig 3A). In contrast, *in vitro* stimulation of bone marrow-derived macrophages, pDCs and conventional DCs (cDCs) lead to the production of proinflammatory cytokines in response to IFNα, but not IFNλ stimulation (Fig 3B). This pattern of response explains findings in corresponding *in vivo* experiments, where naïve mice treated with IFNα had elevated concentrations of proinflammatory cytokines in their BAL fluid at 10 and 18 h post-treatment, whereas neither vehicle control- nor IFNλ-treated mice showed significant increase in these cytokines (Fig 3C). Thus, type I IFN, but not type III IFN, promotes production of proinflammatory cytokines by immune cells, while epithelial cells are not induced to produce these cytokines by either IFN type.

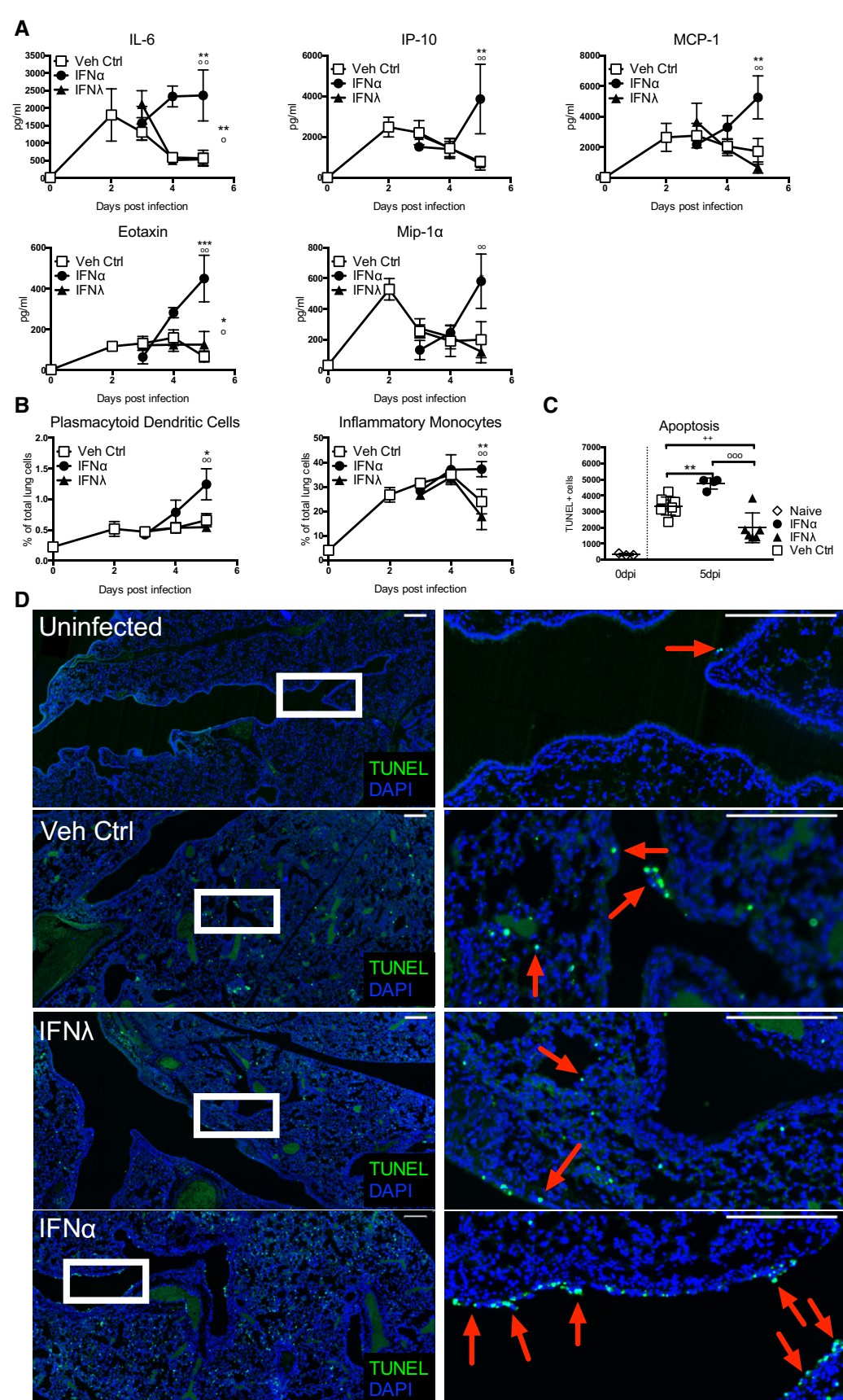

**Figure 2.**

◄

**Figure 2.  IFNα treatment correlates with increased inflammation during IAV infection.**

A, B  Mice were infected with PR8 and treated with IFNα (circles, 1.45 µg/50 µl), IFNλ (triangles, 2.6 µg/50 µl) or Veh Ctrl (squares) as previously stated. Concentrations of stated proinflammatory cytokines in BAL fluid were measured by multiplex cytokine assay (A) and flow cytometric quantification of pDCs and inflammatory monocytes in the lung was performed (B) (data shown are representative of two independent experiments, *n* = 2–6).

C, D  Lung sections from control and infected mice treated as indicated were stained by TUNEL for apoptotic cells at 6 dpi. Quantification of TUNEL[+] cells in whole lung slides by Icy-Spot Detector (ICY-R3M2Y2) (C) (data shown are pooled from three independent experiments, *n* = 3–8). Red arrowheads indicate TUNEL signal (D). Scale bar, 200 µM (data shown are representative of two independent experiments, *n* = 3–4).

Data information: Significance assessed by two-way ANOVA with Bonferroni post-tests (where *denotes IFNα:Veh Ctrl, [+]indicates IFNλ:Veh Ctrl, and °indicates IFNα:IFNλ). Symbols on the right of graphs indicate statistical significance of the whole curve. IL-6 whole curve: **$P = 0.0041$, °$P = 0.0144$. IL-6 5 dpi: **$P = 0.001884$, °°$P = 0.001645$. IP-10 5 dpi: **$P = 0.004897$, °°$P = 0.005354$. MCP-1 5 dpi: **$P = 0.007473$, °°$P = 0.002003$. Eotaxin whole curve: *$P = 0.0235$, °$P = 0.0386$. Eotaxin 5 dpi ***$P = 0.000149$, °°$P = 0.001975$. Mip-1α 5 dpi: °°$P = 0.002921$ (A). Plasmacytoid dendritic cells: *$P = 0.0211$, °°$P = 0.006965$. Inflammatory monocytes **$P = 0.007842$, °°$P = 0.000895$ (B). **$P = 0.0011$, [++]$P = 0.0051$, °°°$P = 0.0005$ (C). Graphs show mean ± SEM.

To identify the overlaps and differences in the lung transcriptional response to IFNα and IFNλ, we performed microarray analysis on whole lungs treated with IFNα, IFNλ or vehicle control (Fig 4). Samples were normalized to the average of the vehicle control group and filtered for a fold change of 1.5, yielding 553 genes differently regulated between treatments, of which 429 genes are upregulated upon treatment. K-means clustering of upregulated genes revealed six gene clusters, one of which contained genes induced by IFNα, but not by IFNλ (IFNα-specific, Fig 4A and Table EV1). The remaining genes were upregulated by both IFNα and IFNλ (Common, Fig 4B and Table EV2). Pathway analysis of the IFNα-specific genes revealed that the top pathways induced are cellular recruitment processes and the "Role of hypercytokinemia/hyperchemokinemia in the pathogenesis of influenza" (Fig 4C), while the pathways common between IFNα and IFNλ were strongly related to IFN signalling pathways, as expected (Fig 4D). This global analysis demonstrated that IFNα has proinflammatory effects not shared with IFNλ, while both IFNα and IFNλ induce canonical antiviral responses in the lung.

**Human primary epithelial and immune cells show the same dichotomy in IFNα versus IFNλ responsiveness as mouse cells**

To translate our results into human cells, we generated human primary AEC cultures and found that both IFNα and IFNλ treatments induced the upregulation of antiviral ISGs such as Rsad2, OAS1 and Mx1 (Fig 5A), as in mouse AEC cultures. PBMCs from healthy donors upregulated ISGs in response to IFNα at both 4 and 24 h (Fig 5B), yet not in response to an equipotent dose of IFNλ. Importantly, analysis of cytokine induction by each IFN treatment at 4 and 24 h revealed that IFNα induced secretion of many proinflammatory cytokines including IL-6, MCP-1 and IP-10, while IFNλ treatment did not (Fig 5C). Collectively, these data indicate that IFNλ treatment of IAV-infected humans is unlikely to drive a cytokine storm and may therefore be a viable treatment option.

# Discussion

An ideal pan-IAV treatment, designed to be given to a population of immunocompetent individuals, should stimulate induction of antiviral genes in AECs to control IAV spread, without driving immunopathology. IFNλ treatment satisfies these criteria, as treatment of IAV-infected mice significantly increased their survival and decreased IAV-induced morbidity. However, treatments of IAV-infected mice with comparable doses of IFNα had the contrary effect, increasing IAV-induced mortality. This highlights the usefulness of IFNλ as a therapeutic against respiratory viruses and underscores the fact that treatment with type I and III IFN can have divergent outcomes, despite inducing a highly similar set of genes in responsive cells (Crotta *et al*, 2013; Lauber *et al*, 2015).

Induction of antiviral factors by either IFNα or IFNλ prior to IAV infection blocks IAV from establishing an infection, thereby inducing sterile immunity and protecting the host. However, this is not a realistic treatment option for a large population. When mouse treatments were started from symptom onset at day 2 of infection both IFNα and IFNλ treatment reduced lung IAV titres compared to infected, vehicle-treated mice. Furthermore, IAV titres in IFNα- and IFNλ-treated lungs were analogous, which indicates equivalent effectiveness of IFNα and IFNλ at restricting IAV replication *in vivo*. IFN therapy administered post-IAV infection likely circumvents the known IAV-induced blockade of these antiviral cytokines. IAV NS1 protein antagonizes induction of both IFNαβ and IFNλ in infected host cells by interfering with upstream pathways such as RIG-I ubiquitination or IRF-3 activation (Gack *et al*, 2007; Hale *et al*, 2008). Exogenous IFN treatment therefore may control IAV spread through the lung by bypassing the block in IFN induction due to IAV NS1 action (Ehrhardt *et al*, 2010) and thus potentiating ISG expression in infected and uninfected cells.

IFNλ controls IAV replication and consequently diminishes infection-induced morbidity. However, why does IFNα treatment translate into increased, not decreased morbidity, despite equivalent control of IAV replication exerted by IFNα as by IFNλ? The explanation for the pathogenicity of the IFNα treatment is increased inflammation through activation and recruitment of immune cells. This hypothesis is supported by increased inflammatory cytokines in BAL fluids, augmented pDC and inflammatory monocyte frequency in the lung and elevated apoptosis of airway epithelial cells following IFNα but not IFNλ treatment. Furthermore, bone marrow-derived macrophages, cDCs and pDCs secreted proinflammatory cytokines *in vitro* only in response to stimulation with IFNα, yet not IFNλ. Importantly, IFNαβ-induced inflammation and AEC apoptosis are immune mechanisms which, when controlled, contribute to limiting the spread of IAV, yet in an immunocompetent system, addition of exogenous IFNα may overactivate these mechanisms resulting in cytokine storm, increased inflammatory cell recruitment, higher frequency of AEC death and ultimately, host mortality. By clustering the response to exogenous IFN treatment into IFNα-specific genes and genes induced by both IFNα and IFNλ,

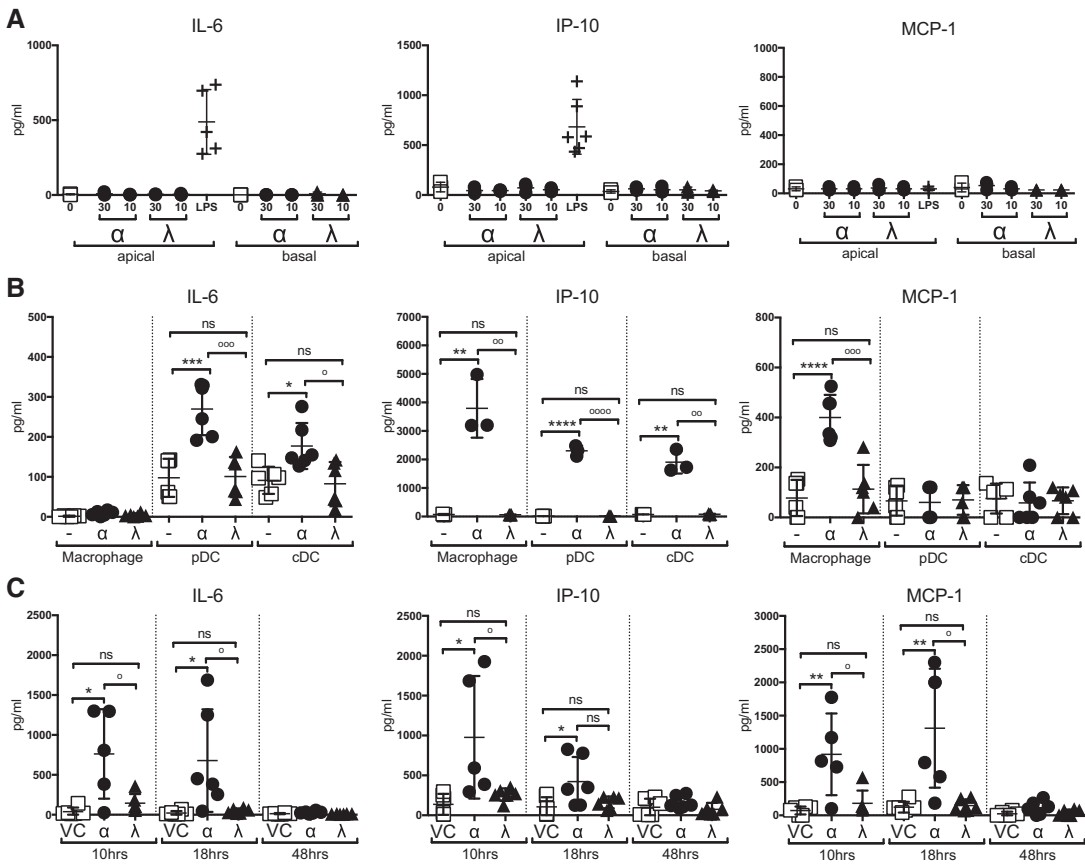

**Figure 3.  IFNα, but not IFNλ, treatment induces pulmonary cytokine secretion through activation of immune cells.**

A, B   IL-6, IP-10 and MCP-1 concentrations were measured by multiplex cytokine assay in AEC culture supernatants (A) and macrophage, pDC and cDC culture supernatants (B) at 24 h post-stimulation with IFNα4 (0.725 ng/ml) or IFNλ2 (1.3 ng/ml) or LPS (AEC only) (data shown are representative of two independent experiments, *n* = 3–6).

C     BAL samples taken from mice treated with IFNα, IFNλ or Veh Ctrl at specified time points (data shown are representative of two independent experiments, *n* = 5–6).

Data information: Significance assessed by unpaired *t*-tests where *denotes IFNα:Veh Ctrl and °indicates IFNα:IFNλ. IFNλ:Veh Ctrl was not significant. IL6 pDC: ***P = 0.0004, °°°P = 0.0005, IL6 cDC: *P = 0.0102, °P = 0.0151. IP-10 macrophage: **P = 0.0033, °°P = 0.0033, pDC: ****P < 0.0001, °°°°P < 0.0001, cDC: **P = 0.0013, °°P = 0.0013, MCP-1 macrophage: ****P < 0.0001, °°°P = 0.003 (B). IL6 10 h: *P = 0.0112, °P = 0.0262, 18 h: *P = 0.0314, °P = 0.373. IP-10 10 h: *P = 0.0261, °P = 0.0472. MCP-1 10 h: **P = 0.0081, °P = 0.0206, 18 h: **P = 0.0089, °P = 0.01 (C). Graphs show mean ± SEM.

we have determined that the pathogenic component driving the hypercytokinemia is specific to IFNα and not common to IFNα and IFNλ. The IFNα-specific gene signature most likely originates from the IFNα-mediated activation of immune cells. This is corroborated both by the lack of production of proinflammatory cytokines by cultured mouse AEC (in this study) and by studies showing a virtually identical gene induction profile between IFNα and IFNλ in cultured murine and human AEC (Crotta *et al*, 2013; Lauber *et al*, 2015).

IAV-induced mortality in humans does not always correlate with high viral load, yet it is always associated with cytokine storm and tissue damage (Peiris *et al*, 2004; de Jong *et al*, 2006; Louie *et al*, 2009; Agrati *et al*, 2010; Arankalle *et al*, 2010). This highlights that AEC apoptosis and therefore lung damage in IAV infection can be caused by both host immunity- and virus-mediated cytotoxicity, and indicates that safe and successful treatment should combat the virus without further immunostimulation.

IFNλ therapy fulfils these criteria as it enhances IAV clearance and thus protects from virus-induced AEC apoptosis, without exacerbating those facets of the immune response that also instigate AEC death.

IFNα treatment drives immunopathology when used therapeutically in the context of a replicating infection. In contrast, prophylactic IFNα treatment is protective, as was shown also in previous influenza studies using mice and ferrets (Kugel *et al*, 2009); what therefore explains this difference? Treatment of uninfected mice with IFNα led to transient proinflammatory cytokine secretion at 10 and 18 h post-infection, yet these cytokines were no longer detected at 48 h post-treatment, and no immunopathology was observed. Moreover, IFNα pretreatment blocked virus infection from the onset leading to sterile immunity, and in this context IFNα-induced inflammation did not augment disease burden. Thus, IFNα alone, in the absence of a further driver of inflammation such as IAV, is insufficient to bring about immunopathology.

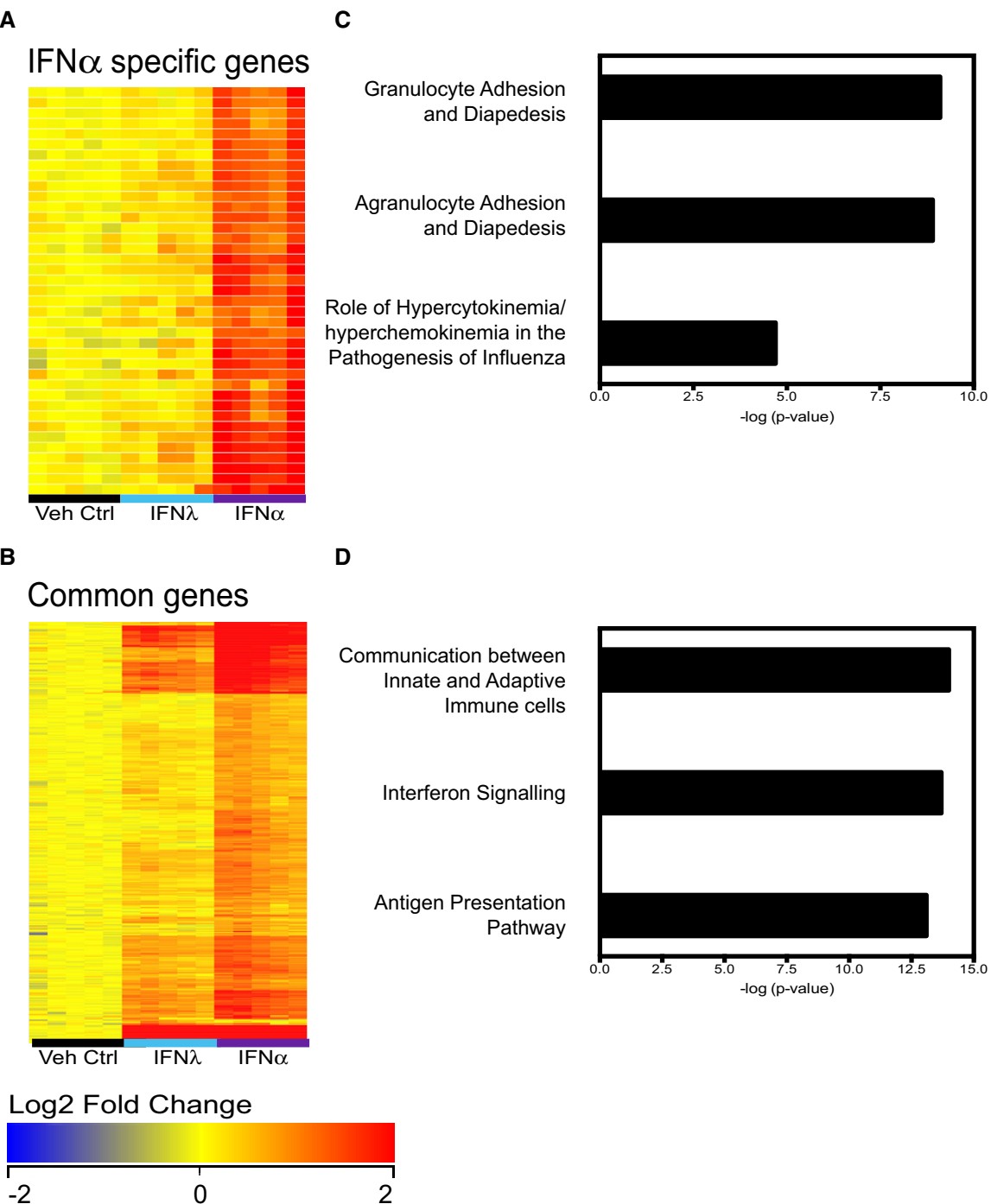

**Figure 4. Pathogenicity-related gene clusters are specifically induced by IFNα, not by IFNλ treatment.**

A–D  Mice were treated with IFNα (1.45 µg/50 µl), IFNλ (2.6 µg/50 µl) or Veh Ctrl (50 µl PBS), and whole lungs were taken at 18 h post-treatment for global analysis by Illumina.SingleColor.Mouse WG-6V20R01127 microarrays. Samples (*n* = 5) were normalized to the median of the vehicle control group and filtered for a fold change of 1.5, yielding 553 genes differently regulated between treatments (one-way ANOVA, *P* < 0.01, Benjamini–Hochberg multiple test correction), of which 429 genes are upregulated. K-means clustering revealed six gene clusters, one of which encompassed genes primarily induced by IFNα4 (A), while the remaining clusters contained genes upregulated by both IFNα4 and IFNλ2 (B). The two clusters of genes were analysed by Ingenuity Pathway Analysis (IPA) (C, D).

This may also explain why in another study, IFNα treatment 1 h after i.p. infection with encephalomyocarditis virus was protective, as this time window may be early enough to abort infection (Tovey & Maury, 1999). In contrast, when added to an ongoing immune response driven by a replicating pathogen such as IAV, IFNα treatment can exacerbate immunopathology. We therefore conclude that IFNαβ-based therapies are not appropriate for IAV treatment.

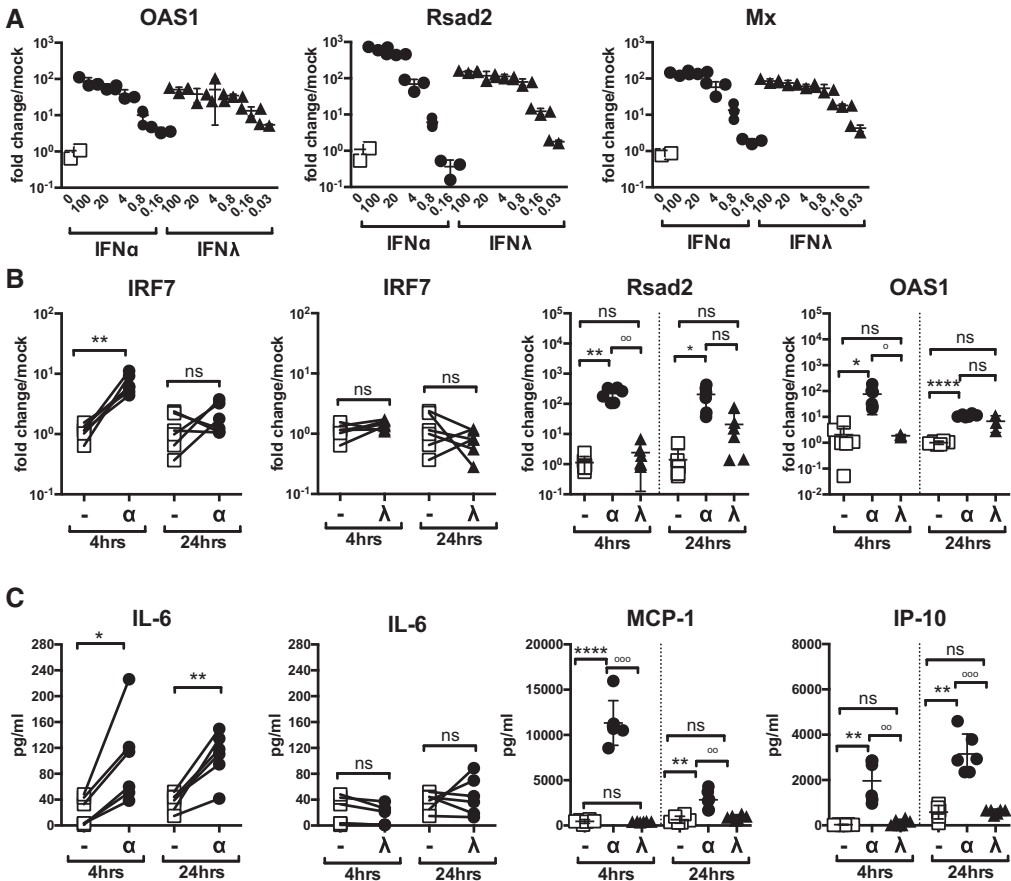

**Figure 5. IFNα, but not IFNλ, treatment induces cytokine secretion from human immune cells.**

A    Human AEC cultures were stimulated for 4 h with IFNα (circles) or IFNλ (triangles) at specified concentrations and then assessed for stated ISG induction by qPCR (data are representative of 2 independent experiments, $n = 3$).

B, C  ISG induction in human PBMCs was assessed at 4 and 24 h post-IFNα (21 U/ml) or IFNλ (1.2 ng/ml) stimulation (B). PBMC proinflammatory cytokine secretion was measured by multiplex cytokine assay at 4 and 24 h post-stimulation with IFNα or IFNλ (C) (data shown are pooled from six independent donors).

Data information: Significance tested by two-way ANOVA where *denotes IFNα:Veh Ctrl and °indicates IFNα:IFNλ. IFNλ:Veh Ctrl was not significant. IRF7: **$P = 0.0037$. Rsad2 4 h: **$P = 0.0038$, °°$P = 0.037$; 24 h: *$P = 0.0234$. OAS1 4 h: *$P = 0.0328$, °$P = 0.0358$; 24 h: ****$P < 0.0001$ (B). IL-6: *$P = 0.0124$, **$P = 0.0021$. MCP-1 4 h: ****$P < 0.0001$; °°°$P = 0.001$; 24 h: **$P = 0.0046$, °°$P = 0.005$. IP-10 4 h: **$P = 0.0033$, °°$P = 0.0024$; 24 h: **$P = 0.0011$, °°°$P = 0.001$ (C). Graphs show mean ± SEM.

Our results are in line with previous studies of severe IAV infection that place endogenous IFNαβ upstream of immune-mediated AEC apoptosis and subsequent host morbidity (Hogner et al, 2013; Davidson et al, 2014). However, the pathogenic potential of IFNαβ reported in these studies was revealed in mouse strains that do not express a functional Mx1 protein. This is particularly significant in terms of translation into human IAV infection as the human homologue of Mx1, MxA, has also been shown to restrict IAV both in vitro and in vivo (Pavlovic et al, 1992, 1995). The mice used in this study may therefore represent a more appropriate model for human IAV infection. While it has to be confirmed whether influenza-infected ferrets or indeed humans show similar responses to IFN treatment, this is to our knowledge the first time that IFNαβ-driven immunopathology has been demonstrated to outweigh the protective effect of the IFN-induced potent anti-IAV protein Mx1 (Horisberger et al, 1983; Salomon et al, 2007).

Unlike the ubiquitously expressed IFNαβR, IFNλR1 is restricted primarily to mucosal surfaces such as the lung epithelial layer

(Sheppard et al, 2003; Mordstein et al, 2008; Sommereyns et al, 2008; Pott et al, 2011). Infection of IFNαβR/IFNλR double-deficient mice with a panel of respiratory pathogens including IAV revealed that the lungs of these mice were highly permissive to viral replication. IFNαβR$^{-/-}$ IFNλR$^{-/-}$ mice had significantly higher pulmonary titres of IAV compared not only to wild-type mice, but also to mice singly deficient for IFNαβR or IFNλR. Increased virus load in IFNαβR$^{-/-}$ IFNλR$^{-/-}$ mice correlated with higher disease burden and host mortality (Mordstein et al, 2008). A further study confirmed that signalling of IFNαβR and IFNλR is entirely redundant in AECs and genetic ablation of both IFNαβR and IFNλR specifically in AECs in vivo resulted in high IAV loads and host morbidity and mortality, in spite of a wild-type immune system (Crotta et al, 2013). While these studies demonstrate that IFNαβ and IFNλ have redundant roles in the control of IAV replication, this is not the case for all viruses. Pretreatment with IFNλ did not alter hepatotropic virus-induced disease progression, and IFNαβR$^{-/-}$ mice were just as susceptible to these viruses as IFNαβR$^{-/-}$ IFNλR$^{-/-}$ mice (Mordstein et al, 2008). Thus, the effectiveness of IFN treatment to

ameliorate virally induced disease is intimately linked to virus tissue tropism. We demonstrate here that due to restricted cellular distribution of IFNλR, which overlaps with IAV tissue tropism, IFNλ treatment promotes antiviral gene induction in relevant cell types without increasing immune cell stimulation, making IFNλ an attractive therapy for clinical IAV-induced disease. As many respiratory viruses exhibit the same epithelial tissue tropism as IAV, IFNλ may be a promising therapeutic for other respiratory viruses such as new emerging corona viruses.

We replicated our data on human AEC cultures and human PBMCs. As expected, both IFNα and IFNλ induce ISG expression in AEC cultures, but only IFNα induces cytokine secretion from human PBMCs. Thus, the *in vitro* and *in vivo* data presented in this paper suggest that IFNλ does not stimulate or modulate mouse or human immune cell function in this disease model; however, there is emerging evidence that in some instances IFNλ-mediated immunomodulation may occur (Jordan *et al*, 2007; Ank *et al*, 2008; Dai *et al*, 2009; Liu *et al*, 2011; Egli *et al*, 2014a; Blazek *et al*, 2015; de Groen *et al*, 2015a). Among these IFNλ effects are the reduction in neutrophil numbers and IL-1β production in chronic Th17-driven disease (Blazek *et al*, 2015) and suppression of Th2 cytokines (Dai *et al*, 2009; Koltsida *et al*, 2011. These effects can also be mediated by IFNα (Moro *et al*, 2016, reviewed in Davidson *et al*, 2015) and are most likely of limited importance in our infection model with low neutrophil numbers and IL-1β and Th2 cytokine levels. In addition, we find no significant changes in neutrophil numbers by either IFN treatment in infected mice.

IFNλ-dependent enhanced production of IL-12 in human macrophages (but not monocytes or DCs) and of IFNγ by human NK cells has also been described (Liu *et al*, 2011; de Groen *et al*, 2015a), but we find that IFN treatments did not change IFNγ and IL-12 levels in infected lungs. Ank *et al* (2008) find that only a restricted set of cells, including pDCs and epithelial cells, can be activated *in vitro* by IFNλ, while other immune cells are not activated by IFNλ. Our findings are in agreement with their results for epithelia and other immune cells tested; however, we find no cytokine production by Flt3-generated BM-derived pDCs *in vitro*. Since their pDCs are purified *ex vivo*, the discrepant findings may be explained by this technical difference. Finally, diverging results have been found for B cells, with some studies proposing that a high IFNλ environment suppresses human B-cell responses, while others show *in vitro* that IFNλ, like IFNα, can promote human B-cell activation (Egli *et al*, 2014a,b; de Groen *et al*, 2015b). We find no changes in B-cell recruitment to the lung by either IFN treatment, but enhanced B-cell activation mediated only by IFNα, not IFNλ, again underlining the lack of direct immune cell activation by IFNλ we observe in our model.

In an adoptive transfer model of anti-tumour immunity, it was shown that NK cells require the IFNλ receptor IL-28R for full activity (Souza-Fonseca-Guimaraes *et al*, 2015). The authors also describe that effects of IFNα and IFNλ are additive, suggesting that these IFNs have a similar role in NK cell activation. It is therefore unlikely that NK cell actions explain the differential effects of IFNα versus IFNλ in our model.

Further clinical research will be required to compare the proinflammatory potential of type I and type III IFNs in human IAV. Studies in HCV patients indicated a higher incidence of liver-related adverse events during IFNλ versus IFNα treatment,

but lower haematological abnormalities such as neutropenia or thrombocytopenia and fewer influenza-like symptoms such as pyrexia, chills or pain in patients treated with IFNλ compared to IFNα, in line with lower direct immunomodulatory effect of IFNλ (Muir *et al*, 2014). Based on the immunosuppressive effect of IFNλ on neutrophils (Blazek *et al*, 2015), clinical trials are also planned to test whether IFNλ can reduce neutrophil-mediated pathology in rheumatic disease (https://clinicaltrials.gov/ct2/show/NCT02498808). An additional complexity is that both endogenous and administered IFNs may interact in an additive, synergetic or cross-inhibitory manner, as demonstrated for the interplay between IFNα and IFNγ (Lasfar *et al*, 2014) and the intricate relationship between IFNα and IFNλ in hepatitis C (Duong *et al*, 2014; Egli *et al*, 2014b). Since mice, like humans, contain IFNλ-responsive immune cells (Koltsida *et al*, 2011; Blazek *et al*, 2015), it is encouraging that IFNλ treatment does not contribute to a "cytokine storm" or host pathology during IAV infection, as we demonstrated here. We suggest therefore that IFNλ should be the preferred antiviral compound for treatment of IAV infections.

As the human population expands, the interface between the animal reservoir of IAV and the human population grows. Increased contact increases the likelihood of a novel IAV strain to cross the species barrier. Coupled with the difficulty of developing an IAV vaccine that will induce broad protection, the time lag between vaccination and host protection, and IAV's exceptional ability to escape host adaptive immunity through antigenic shift and drift, development of a treatment that will stimulate protective aspects of the host immune response to IAV is highly desirable. Furthermore, targeting IAV directly drives the emergence of drug-resistant strains due to the high natural mutation rate of IAV. As IAV has already evolved mechanisms to antagonize the induction of IFN and given the multiplicity of antiviral effectors that are induced by IFNs, addition of more of these cytokines to an infected system may serve to circumvent IAV-mediated block of IFNs while also making it difficult for the virus to evolve mutants to escape such a multifaceted antiviral response. In contrast to the ubiquitous effects of IFNα, the match of IFNλR expression and IAV tissue tropism allows IFNλs to target cell types at risk of infection, effectively inducing antiviral genes in these cells and therefore assisting in the control of IAV spread, without the risk of stimulating the immune system to enhance pathology.

## Materials and Methods

### Ethics statement

All protocols for breeding and experiments with animals were approved by the local ethics committee of the Francis Crick Institute, Mill Hill Laboratory (FCI-MH), and by the Home Office, UK, under the Animals (Scientific Procedures) Act 1986 and project licence 70/7643. For *in vitro* experiments with human immune cells, peripheral venous blood was obtained from six healthy adult volunteers. A properly executed, written, and FCI-MH review board-approved informed consent was obtained from each volunteer before blood collection. All samples were collected according to protocols approved by the FCI-MH.

## Mice

All experiments used 6- to 12-week-old B6.A2G-Mx1 congenic mice carrying functional Mx1 alleles on the C57BL/6 background (Staeheli et al, 1985) (kind gift from Dr P. Staeheli, Freiburg Univ.), (B6.A2G-Mx1 × C57BL/6)F1 mice, or control C57BL/6 mice bred at the FCI-MH under specific pathogen-free conditions.

## Recombinant IFN proteins

A codon-optimized cDNA encoding the mature form (without the signal peptide) of mouse IFNλ2 was purchased (Eurofins) and expressed in *E. coli*, purified under denaturizing condition and refolded *in vitro* as described previously (Dellgren et al, 2009). The human IFNλ3 was made as described in Dellgren et al (2009). Mammalian IFNα4 for mouse studies and human universal type I IFN were purchased from PBL Assay Science.

## Primary mouse tracheal epithelial cell culture

Isolation and culture of primary mouse airway epithelial cell culture (AEC) were performed as previously described (Crotta et al, 2013): in brief, cells were isolated from C57BL/6 mouse trachea by enzymatic treatment and seeded onto a 0.4-μm pore size clear polyester membrane (Corning) coated with a collagen solution. At confluence, medium was removed from the upper chamber to establish an air–liquid interface (ALI). Fully differentiated, 7- to 10-day-old post-ALI cultures were routinely used for experiments. For analysis of cytokine secretion, AEC cultures were stimulated with IFNα (0.725 ng/ml), IFNλ (1.3 ng/ml) or LPS (1 μg/ml, Lonza) or medium control for 24 h. Supernatants were then collected and stored at −70°C until samples were analysed.

## IFN titration on AEC and subsequent generation of an IFNα:IFNλ conversion ratio

AEC cultures were stimulated for 4 h with serial dilutions of IFNα or IFNλ or medium control ranging from 2.175 to 0.022 ng/ml for IFNα and from 6 to 0.003 ng/ml for IFNλ, and induction of stated ISGs was assessed by qPCR. For each gene, data were pooled from two independent IFNα and IFNλ titrations. Prism 6 software was used for four-parameter logistic regression analysis, to generate a dose–response curve and obtain half-maximal effective concentrations ($EC_{50}$) for each gene assessed for each treatment. An IFNα:IFNλ conversion ratio was then generated by dividing the IFNα $EC_{50}$ for an ISG by the IFNλ $EC_{50}$ for the same gene. The final conversion ratio of 0.558 was determined by the geometric mean of the ratios obtained for all ISGs assessed and applied to treat mice with equipotent amounts of IFNα and IFNλ.

## Infection and treatment of mice

Influenza A virus (strain A/Puerto Rico/8/34 H1N1) (PR8) (kind gift from Dr J. Skehel, FCI-MH) was grown in Madin-Darby canine kidney (MDCK) cells, a kind gift from Dr J. McCauley, FCI-MH, stored at −70°C and titrated on MDCK cells by 50% tissue culture infective dose ($TCID_{50}$), according to the Spearman-Karber method. B6.A2G-Mx mice were infected with PR8

$(3 \times 10^4 - 1 \times 10^5 TCID_{50}/30\ \mu l)$. B6.A2G-Mx mice were treated with 1.45 μg/50 μl of IFNα or 2.6 μg/50 μl IFNλ either at −1 dpi (pretreatment experiment) or days 2, 4 and 5 post-infection (treatment during infection experiments). Mice were infected and treated under light isoflurane-induced anaesthesia intranasally. All anaesthesia was performed with animals kept on a heat mat to regulate body temperature.

## *In vitro* stimulation of pDCs, cDCs and macrophages

C57BL/6 bone marrow cells were obtained by crushing femurs and tibias with a mortar and pestle in RPMI-1640 (BioWhittaker). Red blood cells were lysed using ammonium chloride, and cells were cultured in culture media (10% foetal calf serum (PAA), L-glutamine, penicillin, streptomycin and β-mercaptoethanol in RPMI-1640) supplemented with Flt3L (100 ng/ml, PeproTech) for pDCs and cDCs or, for macrophages, supplemented with L cell sup (10%, kind gift from Anne O'Garra, FCI-MH) culture media. Media were replaced at day 4 of cultures and harvested at day 7. Macrophages were isolated from culture by collection of the adherent cells. Culture was found to contain 95% macrophages, identified as $FSC^{hi}$, $SSC^{hi}$, $F4/80^+$, $CD11b^+$ by flow cytometry. For pDCs and cDCs, non-adherent cells were collected and pre-incubated with Fc blocking mAbs and biotin-conjugated anti-B220 (clone RA3-6B2, BioLegend) in 2% FCS (PBS) before a 30-min incubation with anti-biotin-conjugated magnetic beads. pDCs were then positively selected using LS columns and the QuadroMACS separator, following the manufacturer's instructions (Miltenyi Biotec), and found to be 95% pure based on $FSC^{lo}$, $SSC^{lo}$, $PDCA-1^+$ and $Siglec-H^+$ as analysed by flow cytometry. cDCs were collected from negative fraction and were found to be 90% pure based on $FSC^{int}$, $SSC^{int}$, $CD11c^+$ and $CD11b^+$. All cell types were seeded at $2 \times 10^5$ cells per well and rested for 24 h before stimulation with IFNα (0.725 ng/ml), IFNλ (1.3 ng/ml) or media controls for 24 h. Supernatants were then collected and stored at −70°C until samples were analysed.

## Stimulation of human cells

The human biological samples were sourced ethically and their research use was in accord with the terms of the informed consents. Primary human bronchial epithelial cells were purchased from Lonza and cultured as per manufacturer's instructions. In brief, cells were expanded in a T-75 flask to 60% confluence and then harvested for seeding onto transwells at 50,000 cells per insert. At confluence, liquid was removed from the upper chamber to establish ALI. Fully differentiated, 15- to 20-day-old post-ALI cultures were routinely used for experiments. Peripheral blood mononuclear cells (PBMCs) were prepared from peripheral blood by Ficoll-Paque density gradient centrifugation and cultured at $2 \times 10^5$ cells per well. For analysis of cytokine secretion, primary human bronchial epithelial cell and PBMC cultures were stimulated with human universal type I IFN (21 U/ml) and human IFNλ (1.2 ng/ml) or media controls for 24 h.

## Viral quantification by $TCID_{50}$

Whole lungs from infected mice were collected on ice. Lungs were minced and pressed through a 70-μM strainer using 1 ml of PBS.

Samples were then centrifuged at 1,300 rpm, 5 min at 4°C and supernatant collected, stored at −70°C until analysed. All samples within an experiment were titrated on the same day on MDCK cells by $TCID_{50}$, calculated according to the Spearman-Karber method.

## Protein analysis

BAL fluid was recovered from naïve, infected and/or treated mice at specified time points, centrifuged at 1,300 rpm, 5 min at 4°C and supernatant collected. IFN-stimulated mouse pDC, cDC, macrophage and AEC supernatants and IFN or virus-stimulated human PBMC and AEC supernatants were collected after 24 h stimulation. Concentrations of IL-6, IP-10, MCP-1, eotaxin and Mip-1α were assessed by using either a mouse or human multiplex Milliplex Map Kit (Millipore) as per the manufacturer's instructions, and analysed on a Luminex 100 (Bio-Rad).

## Flow cytometry

Leucocytes from the lung were enumerated using flow cytometry. In brief, lungs were excised from naïve, infected and/or treated mice, digested with 20 μg/ml Liberase TL (Roche) and 50 μg/ml DNAse I (30 min at 37°C) and homogenized using gentleMACS (Miltenyi), following the manufacturer's instructions. Lungs were then passed through a 70-μM cell strainer and washed with FACS buffer (10% BSA in PBS azide). Red blood cells were lysed using ammonium chloride, and cells were seeded into a 96-well U-bottom plate at $2 \times 10^6$ per well. Cells were pre-incubated with anti-FcγRIII/II (Fc blocking) mAbs in FACS buffer before a 30-min incubation with one or more of the following fluorochrome-labelled antibodies (purchased from BioLegend and used 1:100, unless otherwise indicated): FITC-conjugated PDCA-1 (clone 927), PE-conjugated Siglec-H (clone 551), PerCP Cy5.5-conjugated Ly6C (clone HK1.4), PE Cy7-conjugated CD11b (clone M1/70), APC-conjugated CD45 (clone 30-F11), APC-conjugated F4/80 (clone BM8), Brilliant Violet 450-conjugated CD11c (clone N418), APC-conjugated CD4 (clone GK1.5, 1:400), PerCP Cy5.5-conjugated CD8 (clone 53-6.7), AF700-conjugated CD3 (clone 17A2), BV650-conjugated CD19 (clone 6D5), PE-conjugated NKp46 (clone 29A1.4), PE-Cy7-conjugated NK1.1 (clone PK136), FITC- or PeCy7-conjugated CD69 (clone H1.2F3), AF700-conjugated Ly6G (clone 1A8), and counterstained with Zombie Aqua (BioLegend) to enumerate dead cells. All samples were resuspended in PBS and analysed using a BD LSRFortessa X-20 (Becton Dickinson).

## Histology

Whole lungs were perfused with 10% neutral buffered formaldehyde *in situ*. Tissue was then fixed overnight in 10% neutral buffered formaldehyde, embedded in paraffin and sectioned. For TUNEL staining, slides were deparaffinized and stained for apoptotic cells using ApopTag Fluorescein *In Situ* Apoptosis Detection Kit (Miltenyi) as per the manufacturer's instructions. Imaging of slides was performed on a VS120 slide scanner (Olympus) with a VC50 camera, a UPLSAPO lens, at magnification of 20× and a numerical aperture of 0.75. Images were analysed using OlyVia Image Viewer 2.6 (Olympus). Quantification of TUNEL-positive cells in whole lung sections was then performed using Icy-Spot Detector.

## RNA extraction

Whole lungs were collected in TRIzol (Invitrogen) and homogenized using Polytron PT 10–35 GT (Kinematica). Mouse or human AEC cultures were lysed directly in the transwells and PBMC cultures were lysed directly in wells using the Qiagen RNeasy mini kit, according to the manufacturer's instructions. Total RNA was prepared using phenol/chloroform extraction, and cDNA was generated from these samples using Thermoscript RT–PCR system, following manufacturer's instructions (Invitrogen). The cDNA served as a template for the amplification of genes of interest and the housekeeping gene Hprt1 by real-time PCR, using TaqMan Gene Expression Assays (Applied Biosystems), Universal PCR Master Mix (Applied Biosystems) and the ABI-PRISM 7900 sequence detection system (Applied Biosystems). The fold increase in mRNA expression was determined using the $\Delta\Delta C_t$ method relatively to the values in mock-treated samples, after normalization to Hprt1 gene expression.

## Microarray data analysis

Lungs were homogenized in TRI Reagent (RiboPure kit, Ambion) and total RNA isolated according to manufacturer's instructions. RNA was hybridized to Illumina.SingleColor.Mouse WG-6_V2_0_R0_1127 microarrays. Raw data were processed using GeneSpring GX version 11.5 (Agilent Technologies). After background subtraction, each probe was attributed a flag to denote its signal intensity detection *P*-value. Flags were used to filter out probe sets that did not result in a "present" or "marginal" call in at least 50% of the samples, in any one out of three experimental conditions. The signal intensity of each probe was first normalized on the median intensity of that probe across the control group and then represented as log2 fold change relative to the controls. Subsequent statistical analysis was a one-way ANOVA to identify genes significantly differentially expressed relative to controls (fold change of ≥ 1.5; $P < 0.01$, Benjamini–Hochberg multiple test correction) which were further analysed by K-means clustering. Microarray data have been deposited in Gene Expression Omnibus database under accession code GSE70628.

## Statistical analysis

Data are shown as the means ± SEM. Sample sizes were designed to give statistical power, while minimizing animal use. For *in vivo* experiments, treatment groups of $n = 6$ were standard, and results from several experiments were pooled to reach statistical significance. In *in vivo* time course experiments, biological triplicates were chosen as minimum. As a pre-established criterion, animals with no weight loss (weight at all times > 98% starting weight) during influenza infection were excluded as not infected. Mice from the same litter and co-housed were allocated to different treatment groups prior to start of experiment, to avoid subjective bias of allocating mice into treatment groups after symptom onset. Data sets were analysed by two-way ANOVA with Bonferroni post-tests (weight, cytokine concentration and cellular recruitment time courses), log-rank (Mantel–Cox) test (survival) and Student's *t*-tests (cytokine or gene induction from cells) or two-way ANOVA (human samples). GraphPad Prism 6 (GraphPad Software, San Diego, CA) was used

**The paper explained**

**Problem**

Influenza is a disease caused by the infection of lung tissue with influenza virus. Lung damage occurring during severe influenza can be inflicted either by the virus itself or by the antiviral immune response. Interferons are a family of soluble molecules called cytokines which are able to induce an antiviral programme in cells and thereby protect these cells from infection. Therefore, interferons are frequently considered as universal anti-influenza treatment. Interferons can also stimulate immune cells, which can be beneficial or damaging in influenza infection. Different types of interferons have different effects on immune cells and thereby contribute differently to enhancing the immune response during influenza infection.

**Results**

Here, we showed that interferon alpha (IFNα) and lambda (IFNλ) were both efficient at inducing the antiviral programme in airway epithelia, but only IFNα activated immune cells. When IFNα or IFNλ was used to treat influenza-infected animals, both were able to reduce virus load, but IFNα also increased the ongoing immune response leading to inflammation and lung damage. Therefore, IFNλ treatment protected because it helped control the virus without increasing lung inflammation, while IFNα aggravated disease because it enhanced the immune-mediated lung damage.

**Impact**

Understanding the multiple effects of interferons is important when considering them as therapeutic options against influenza.

for data analysis and preparation of all graphs. *P*-values < 0.05 were considered to be statistically significant.

**Expanded View** for this article is available online.

## Acknowledgements

We thank A. O'Garra, G. Kassiotis, M. Wilson, K. Bradley, P. Staeheli, J. Skehel, J. McCauley and B. Stockinger for reagents, support and advice. We are grateful to Biological Services at Mill Hill for excellent animal husbandry and to the FACS, Histology, Advanced Sequencing and Microscopy facilities at Mill Hill for outstanding technical support. This study was supported by the Francis Crick Institute which receives its core funding from Cancer Research UK (FC001206), the UK Medical Research Council (FC001206) and the Wellcome Trust (FC001206). Support by MRC grant U117597139 (SD, TMM, SC, AW), a BBSRC-GSK-funded case studentship BB/K501475/1 (TMM) and Danish Council for Independent Research, Medical Research (grant 11-107588, HHG, RH) is gratefully acknowledged.

## Author contributions

SD and AW conceived study; SD, TMM, SC, HHG performed experiments; SD, TMM, SC, EMH, SB, RH and AW analysed and interpreted data; SD, TMM and AW wrote the manuscript; SC, HHG, EMH, SB and RH contributed to manuscript writing; and HHG and RH provided vital reagents.

## Conflict of interest

EMH and SB are employees of GSK. The other authors declare that they have no conflict of interest.

## For more information

http://www.who.int/topics/influenza/en/

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
