## [Review Process File · EMBO Molecular Medicine]

IFN λ is a Potent Anti-Influenza Therapeutic without the Inflammatory Side Effects of IFN α Treatment

Sophia Davidson, Teresa McCabe, Stefania Crotta, Hans Henrik Gad, Edith Hessel, Soren Beinke, Rune Hartmann, Andreas Wack

Corresponding author: Andreas Wack, Francis Crick Institute

Review timeline:

Submission date:	16 March 2016
Editorial Decision:	13 April 2016
Revision received:	01 June 2016
Editorial Decision:	28 June 2016
Revision received:	08 July 2016
Accepted:	21 July 2016

Transaction Report:

Editor: Celine Carret

1st Editorial Decision

13 April 2016

Thank you for the submission of your manuscript to EMBO Molecular Medicine. We have now heard back from the three referees whom we asked to evaluate your manuscript. Although the referees find the study to be of potential interest, they also raise a number of concerns that need to be fully addressed in the next final version of your article.

You will see that while referee 2 is mainly supportive, the other two referees are much more reserved. Referee 1 is concerned about the animal model used and would have appreciated a second animal model to increase the clinical advance of the study. Referee 3 requests more insights on type III IFN mechanisms, which to some extent referee 1 and an additional editorial board adviser we consulted with, agreed to. In essence, while we will not ask you to repeat the therapeutic experiments in vivo in ferrets, we would like you to address the mechanistic requests as much as you possibly can. For example (but this is by no means exhaustive), the microarray data should be compared to existing datasets in order to derive insights from lungs vs. single cells. Literature regarding IFN λ should be more thoroughly checked and appropriately discussed, especially as clinical studies do exist, some side effects were reported and IFN λ responses by neutrophils and NK cells have recently been reported.

Overall, I would like to give you the opportunity to revise your manuscript, with the understanding that the referees' concerns must be fully addressed and that acceptance of the manuscript would entail a second round of review. Please note that it is EMBO Molecular Medicine policy to allow only a single round of revision and that, as acceptance or rejection of the manuscript will depend on another round of review, your responses should be as complete as possible.

Please read below for important editorial formatting.

I look forward to receiving your revised manuscript.

***** Reviewer's comments *****

Referee #1 (Comments on Novelty/Model System):

Not surprising according to previous literature. The use of a second animal model for influenza, the ferret, would increase the chances for a potential clinical application of this

Referee #1 (Remarks):

This is a well conducted study where the authors show that treatment of mice with type I or type II IFN prior to influenza virus infection results in protection from disease; however, treatment after influenza virus infection resulted in no protection from disease (and perhaps a subtle disease enhancement) when type I IFN was used. By contrast, treatment with type III IFN protected from disease. This was not due to higher antiviral activity of type III versus type I IFN, but to higher pro-inflammatory properties of type I IFN. Some other ancillary experiments are conducted in this report, such as treatment of epithelial cells or microarrays, but these are not essential for the message, and do not add significant novel information. While the conclusions are clear, there are perhaps predictable according to previous literature. The same group has previously demonstrated that type I IFN treatment induces immunopathology during influenza virus infection, and it is known that type III IFN is signaling in epithelial cells, but not in immune cells, and therefore, induces fewer side effects. The main difference here is that the authors use now Mx1 mice, which mirrors better the situation in humans, where Mx1 is also expressed in response to IFNs. However, this is a very small incremental advance. The study could have been perhaps more relevant if in addition to show these results in the Mx1 mouse model, the authors would have show these results in the ferret model. Kugel et al administered type I IFN to influenza virus infected ferrets and did not see any detrimental effect (J. Virol. April 2009 vol. 83 no. 8 3843-3851). A comparison of type I IFN with type III IFN in a different and relevant animal model for influenza, such as the ferret model, would enhance the novelty of this report.

Referee #2 (Comments on Novelty/Model System):

IFN λ binds to receptors expressed on a limited array of cell types, notably cells of epithelial origin. Here the authors have used a mouse model and human cells cultured in vitro to demonstrate that IFN λ activates expression of an antiviral program in airway epithelial cells without induction of inflammatory cytokines and chemokines by blood cells. In mice, IFN λ treated established influenza A virus infection while IFN α treatment enhanced pathogenicity and death. The paper is largely well-written, quite complete-incorporating gene expression, some histology, some (summarized) flow cytometric analysis, viral quantitation and cytokine quantitation-and convincing, with clear human application.

Referee #2 (Remarks):

IFN λ binds to receptors expressed on a limited array of cell types, notably cells of epithelial origin. Here the authors have used a mouse model and human cells cultured in vitro to demonstrate that IFN λ activates expression of an antiviral program in airway epithelial cells without induction of inflammatory cytokines and chemokines by blood cells. In mice, IFN λ treated established influenza

A virus infection while IFN α treatment enhanced pathogenicity and death. The paper is largely well-written, quite complete-incorporating gene expression, some histology, some (summarized) flow cytometric analysis, viral quantitation and cytokine quantitation-and convincing, with clear human application. I have a few comments.

1. In the methods (e.g. lines 344, 349) the authors use ng/ml to describe IFN concentrations. In Figure EV1, they use U/ml. In figures 1 and 5 the units are not specified. Please clarify.
2. Cultured tracheal/bronchial epithelial cells are abbreviated as AEC (line 338; I assume the A is for Airway) and this abbreviation is used throughout the manuscript. Yet on line 408-and nowhere else in the manuscript-the authors use the abbreviations mTEC and hTEC. T for tracheal?
3. On lines 429-434, the authors provide methods for intracellular staining for viral NP/M proteins. No data are shown or mentioned for this experiment.
4. In the legend to Figure 1 (line 653-654), the phrase "IFN λ :Veh Ctrl was not significant" may have been duplicated from the legend for figure 2. In Figure 1, there are many measurements (survival, viral load, body weight) for which there is indeed a significant difference between IFN λ and the vehicle control.
5. The authors refer repeatedly to measuring cytokines by "multiplex." Perhaps this could be revised to provide a little more information (e.g. Luminex, other multiplex assays).
6. Figure 4 legend, line 689: Do you mean (C,D) rather than (D,E)?
7. Figure 5A. The authors refer to 2 independent experiments. Were these experiments performed with two independent AEC donor samples, or from a common lot of cells from a single donor?
8. Figure 5C. "Data is pooled from 6 independent donors." Some of these panels show more than 6 data points. Are some individual donors indicated more than once? Why?

Referee #3 (Remarks):

Major concerns:

The authors present IFN-lambda as not involved in the immune response. However numerous studies have shown that IFN-lambda is a potent immune regulator. The authors should determine the immune regulations of IFN-lambda that are beneficial instead of neglecting them.

The authors need to evaluate the role of endogenous type I and type III IFN on IFN therapy. Prior IFN therapy, virus infected mice should be grouped according to their constitutive expression of type I and type III. Induction of type I and type III IFN by influenza A virus has different impact on the immune response and related effects on IFN therapy and control of infection.

Interpretation of the data based on some effects of exogenous IFN is misleading. The mode of administration of IFN is crucial in IFN dosage and related side effects. Prior studies on IFN-alpha intranasal therapy by Tovey group showed a beneficial effect in virus clearance without any significant exacerbation.

Other concerns:

Title is misleading: not really reflecting the data and the role of IFN λ .

Inconsistent references: The authors need more updates on the advance clinical applications of IFN λ .

1st Revision - authors' response

01 June 2016

***** Reviewer's comments *****

Referee #1 (Comments on Novelty/Model System):

Not surprising according to previous literature. The use of a second animal model for influenza, the ferret, would increase the chances for a potential clinical application of this

Our reply: This is in fact the first study showing that IFNL therapy is protective during influenza infection, and we are able to explain the underlying mechanism and contrast this to IFN α therapy. We therefore think that this finding merits publication in EMBO Mol. Med., even if this outcome may have been a valid prediction given the previous knowledge.

Referee #1 (Remarks):

This is a well-conducted study where the authors show that treatment of mice with type I or type II IFN prior to influenza virus infection results in protection from disease; however, treatment after influenza virus infection resulted in no protection from disease (and perhaps a subtle disease enhancement) when type I IFN was used. By contrast, treatment with type III IFN protected from disease. This was not due to higher antiviral activity of type III versus type I IFN, but to higher pro-inflammatory properties of type I IFN. Some other ancillary experiments are conducted in this report, such as treatment of epithelial cells or microarrays, but these are not essential for the message, and do not add significant novel information. While the conclusions are clear, there are perhaps predictable according to previous literature. The same group has previously demonstrated that type I IFN treatment induces immunopathology during influenza virus infection, and it is known that type III IFN is signaling in epithelial cells, but not in immune cells, and therefore, induces fewer side effects. The main difference here is that the authors use now Mx1 mice, which mirrors better the situation in humans, where Mx1 is also expressed in response to IFNs. However, this is a very small incremental advance. The study could have been perhaps more relevant if in addition to show these results in the Mx1 mouse model, the authors would have show these results in the ferret model.

Our reply: We thank the reviewer for the appreciation of our study and agree with the reviewer that ferret studies are an important next step to assess the clinical potential of our findings. This will require the production of clean ferret IFN α and IFN λ , titration on ferret epithelia to establish equipotency, and treatment of ferrets with equipotent doses, and we will attempt this in a separate study in the future.

Kugel et al administered type I IFN to influenza virus infected ferrets and did not see any detrimental effect (J. Virol. April 2009 vol. 83 no. 8 38433851).

Our reply: We are aware of this study and have now included this in the discussion of our data on page 13, lines 244-245. All ferrets in Kugel et al. are pretreated with IFN α twice prior to infection, and therefore results from that study fit with our data in this study on IFN α pre-treatment where we do not detect detrimental effects by IFN α .

A comparison of type I IFN with type III IFN in a different and relevant animal model for influenza, such as the ferret model, would enhance the novelty of this report.

Our reply: See above, we agree that ferret studies are important and will attempt this in a separate study. The editor agrees that adding ferret studies would go beyond the scope of our study here.

Referee #2 (Comments on Novelty/Model System):

IFN λ binds to receptors expressed on a limited array of cell types, notably cells of epithelial origin. Here the authors have used a mouse model and human cells cultured in vitro to demonstrate that IFN λ activates expression of an antiviral program in airway epithelial cells without induction of inflammatory cytokines and chemokines by blood cells. In mice, IFN λ treated established influenza A virus infection while IFN α treatment enhanced pathogenicity and death. The paper is largely well written, quite complete-incorporating gene expression, some histology, some (summarized) flow cytometric analysis, viral quantitation and cytokine quantitation-and convincing, with clear human application.

Referee #2 (Remarks): IFN λ binds to receptors expressed on a limited array of cell types, notably cells of epithelial origin. Here the authors have used a mouse model and human cells cultured in vitro to demonstrate that IFN λ activates expression of an antiviral program in airway epithelial cells without induction of inflammatory cytokines and chemokines by blood cells. In mice, IFN λ treated established influenza A virus infection while IFN α treatment enhanced pathogenicity and death. The paper is largely well-written, quite complete-incorporating gene expression, some histology, some (summarized) flow cytometric analysis, viral quantitation and cytokine quantitation-and convincing, with clear human application. I have a few comments.

Our reply: Thank you for appreciating our study.

1. In the methods (e.g. lines 344, 349) the authors use ng/ml to describe IFN concentrations. In Figure EV1, they use U/ml. In figures 1 and 5 the units are not specified. Please clarify.

Our reply: We now show throughout the paper mouse IFN concentrations as ng/ml, so fig EV1 is changed, and we have also added now the IFN concentrations in (w/v) in the legends to fig.s 1-3. For hIFNs, we continue to use (U/ml) for IFN α and (ng/ml) for IFN λ and have specified this now in fig.5 and its legend.

2. Cultured tracheal/bronchial epithelial cells are abbreviated as AEC (line 338; I assume the A is for Airway) and this abbreviation is used throughout the manuscript. Yet on line 408-and nowhere else in the manuscript-the authors use the abbreviations mTEC and hTEC. T for tracheal?

Our reply: Thanks for pointing this out. Yes, A stands for airway and T for tracheal, but for consistency we have removed the acronym mTEC entirely and stick to AEC throughout the paper now.

3. On lines 429-434, the authors provide methods for intracellular staining for viral NP/M proteins. No data are shown or mentioned for this experiment.

Our reply: This was our mistake, and we have now removed these phrases from the methods section.

4. In the legend to Figure 1 (line 653-654), the phrase "IFN λ :Veh Ctrl was not significant" may have been duplicated from the legend for figure 2. In Figure 1, there are many measurements (survival, viral load, bodyweight) for which there is indeed a significant difference between IFN λ and the vehicle control.

Our reply: Thanks for pointing out this mistake that we have now corrected in figure 1 and its legend.

5. The authors refer repeatedly to measuring cytokines by "multiplex." Perhaps this could be revised to provide a little more information (e.g. Luminex, other multiplex assays).

Our reply: This assay is explained in the methods section under protein analysis (lines 452-454), and we now refer to these assays in the text as multiplex cytokine assays.

6. Figure 4 legend, line 689: Do you mean (C,D) rather than (D,E)?

Our reply: Yes, thank you. We have corrected this now.

7. Figure 5A. The authors refer to 2 independent experiments. Were these experiments performed with two independent AEC donor samples, or from a common lot of cells from a single donor?

Our reply: We acquired samples from two different lots from Lonza and specifically requested that they would be derived from two different donors. Lonza confirmed this. The two independent experiments were done on the two different lots that we have acquired.

8. Figure 5C. "Data is pooled from 6 independent donors." Some of these panels show more than 6 data points. Are some individual donors indicated more than once? Why?

Our reply: Thanks for pointing this out. Yes by mistake some triplicate measurements of the same donor were left in the analysis, wrongly increasing the sample size above six. We have now replaced these replicates by the mean for that given donor and show the analysis using one value per donor.

Referee #3 (Remarks):

Major concerns: The authors present IFN-lambda as not involved in the immune response. However numerous studies have shown that IFN-lambda is a potent immune regulator.

Our reply: We agree with the referee that studies on direct immunomodulation by IFN λ have been published, and that some of them describe immunoregulation, which, for clarity, we assume the referee intends as effects dampening the immune response (we have also discussed

immunostimulatory effects, in case the referee includes this in the definition). We had already cited many of these studies, including the negative effect of IFNL on neutrophil recruitment and consequently overall IL-1b amounts in collagen-induced arthritis (Blazek et al.), and the reduction of Th2 responses in asthma models (Koltsida et al.). We have now extended the discussion of these papers and added other papers that show direct effects on immune cells (p. 15-16, lines 290-319). Since neither IL-1b nor Th17 nor Th2 responses are predominant in influenza infection, the above studies are important but may not be pertinent to our infection model, as laid out in detail above in the reply to the editor and below.

The authors should determine the immune regulations of IFN-lambda that are beneficial instead of neglecting them.

Our reply: We thank the reviewer for the observation that potentially immunoregulatory effects of IFNL are at work in our system. We have in fact considered this possibility and have reached the following conclusion:

For IFNL-mediated immunoregulatory effects to be relevant to the comparison between IFNa and IFNL treatment performed here, three criteria must be met:

1. The immunoregulatory effect must be specific to IFNL and not exerted by IFNab, otherwise it would not explain the differences between IFNa and IFNL treatments that we observe.
2. The immunoregulatory effect must act on a process that is pathogenic in influenza.
3. The immunoregulatory effect must be detected in our treatment model.

We believe that a number of IFNL-mediated immunoregulatory effects mentioned by the editor and probably referred to by referee 3 do not meet one or several of these criteria, as we will lay out below in detail for the references in question:

IFNL-mediated block of Neutrophil recruitment and NPh IL-1 production.

1. Differential effect of IFNL and IFNa: While there is a detailed description of IFNL-mediated immunoregulation in collagen-induced arthritis (Blazek et al.), the authors of that study themselves refer to several IFNab-mediated effects blocking NPh recruitment and activity. Antagonism between IFNab and IL-1 has been widely covered in the literature, e.g. in influenza-bacterial coinfection, in MTb and in other infections (e.g. Mayer-Barber et al. *Immunity* 2011 (35), 1023–1034; references in Blazek et al.; also reviewed in Davidson et al. *J IFN & Cytokine Res.* 2015 (35), 252-64). Therefore, while IFNL may have antagonistic effects on NPhs and IL-1 production, ample literature suggests that this is not unique to IFNL but shared with IFNab. These effects can therefore not constitute the distinguishing feature between IFNa and IFNL treatment.

2. Pathogenicity of the function targeted by IFNL: The underlying assumption would be that IFNL mediated block of NPh would be protective. We have depleted NPh in influenza infection and did not find any protective or deleterious effect (Davidson et al. *Nat. Comms.* 2014 (5), 3864; Ellis et al. *EMBO Reports*, 2015 (16), 1203–1218). In addition, in our studies on endogenous high IFNab levels (Davidson et al. *Nat. Comms.* 2014 (5), 3864), we find increased NPh levels in IFNAR1 KO (129) mice which show lower influenza severity, again not correlating high NPh numbers with severe influenza. In most influenza infections including ours, IL-1 levels are low, most likely due to the predominance of IFNab and possible IFNL and the antagonisms described. In addition, when the wider literature is considered, IL-1 is overall more protective than pathogenic in influenza throughout a number of studies (Szretter et al. *J Virol* 2007 (81), 2736-2744; Kozak W et al. *Am J Physiol* 1995, (269), R969-977; Schmitz et al. *J Virol* 2005 (79), 6441-6448). Together, our own observations and published data show that reducing NPh numbers or IL-1b in influenza infection is unlikely to reduce pathogenicity.

3. Indications of immune functions suppressed by IFNL treatment: In fact, we do not find IFNL-specific reductions in NPh numbers or in IL-1b levels upon treatment (fig. EV2 and please see fig.3 in this letter above in the response to the editor).

Th2 suppressive effects of IFNL (Koltsida et al., *EMBO Mol. Med.* 2011 (3), 348-61, also Dai et al. *Blood* 2009 (113), 5829-5838).

1. Differential effect of IFNL and IFNa: Th2 suppressive effects were also described for IFNab and

IFN γ (Moro Nat Imm 2016 (17), 76-86), and therefore we expect no differential effect between IFN α and IFNL treatments on Th2 cytokine suppression.

2. Pathogenicity of the function targeted by IFNL: Influenza infection primarily induces a Th1 response which contributes to viral clearance. Potential enhancement of Th1 immunity through suppression of Th2 cytokines could be potentially beneficial to IFNL treated mice. However, influenza infection results in very low to no induction of Th2 cytokines so this suppression, if it occurs, may have little consequence to disease outcome.

3. Indications of immune functions suppressed by IFNL treatment: We assessed concentrations of the canonical Th2 cytokines: IL-4, 5 and IL-13 and found comparable levels of both cytokines in all treatment groups. Importantly, these cytokines were observed in very low concentrations. Additionally, we did not observe enhancement of IFN γ secretion in IFNL treated mice, and we therefore conclude that either this effect is not observed here or it occurs at such a low level that it does not impact on disease outcome (see fig. 4,5 in this letter above in the response to the editor).

Increased IL-12 production

It has also been shown that IFNL enhances IL-12 secretion from in vitro TLR-stimulated human macrophages (Liu et al. Blood. 2011 (117), 2385-95). We tested IL-12 levels in BAL fluid and do not observe differences in IL-12 secretion between treatment groups, either during infection (Fig. 6 above) or when mice are treated with IFNs only (Fig. 7 above).

B cell suppressive effects of IFNL (Egli et al. PLoS Pathog 2014 (10): e1004556).

As already quoted in the manuscript, Egli et al. provide evidence by studying individuals with SNPs in IFNL and through in vitro experiments, that lower levels of IFNL in humans associates with better seroconversion in influenza vaccination (Egli et al. Emerging microbes & infections. 2014 (3), e512014) and measles. Egli also shows blockade of IL28Ra with peptide enhances B cell activation in vitro and vaccine response in vivo in human samples. In contrast, a recent report shows that IFNL activates human B cells similarly to IFN α (de Groen, J. Leuk. Biol. 2015 (98), 623). It is therefore unclear whether B cell function should be increased or decreased by IFNL and whether or not there is a differential between IFN α and IFNL. To address this directly, we now include B cell recruitment and activation data and find that IFN α or IFNL do not modulate B cell numbers during influenza infection (Fig. EV2A). Also, CD69 upregulation is strongly enhanced by IFN α but not by IFNL (Fig. EV2B), suggesting that IFN α , but not IFNL, may impact on B cell activation. It is unclear however how this should contribute to IFN α -mediated disease severity. Furthermore, the adaptive immune response is absolutely required for influenza clearance, and IFNL deficient mice achieve this without issue (Mordstein et al., PLoS Pathog. 2008 (4), e1000151). Therefore, if there is a positive or negative effect of IFNL on antibody responses, it does not impact on disease outcome.

Additional immunosuppressive mechanisms potentially at work in our model.

We have assessed the levels of the immunosuppressive cytokine IL-10 and the appearance of FoxP3 expression, to understand if IFN treatment can enhance these. At d5 of treatment, IL-10 levels are increased by both IFN α and IFNL (Fig. 8 below), in agreement with the ample evidence in the literature of the ability of IFN α to induce IL-10. This induction therefore does not distinguish between IFN α and IFNL treatment. It has also been proposed that IFNL specifically promotes FoxP3 expressing Treg cells (Mennechet et al. Blood 2006 (107), 4417-4423). We therefore tested FoxP3 expression in IFN treated lungs and did not find any change (Fig. 9 below).

Fig.8: IL-10 levels in BAL of infected mice with or without IFN treatment:
*Fig.8: Increase of lung IL-10 by both IFNa and IFNL treatment during influenza infection. *, ° $P < 0.05$ where * indicates IFNa:Veh Ctrl, ° indicates IFNL:Veh Ctrl.*

Fig.9: FoxP3 RNA levels in total lung samples from IFN-treated mice.
Fig.9: No change in FoxP3 RNA levels in the lung by either IFNa or IFNL treatment.

To conclude from this exhaustive survey of the literature and from our own data, while we have no doubt about the importance of IFNL-mediated immunoregulatory or immunostimulatory effects, we find no evidence that they are at work in our model, or that they explain the difference between IFNa and IFNL treatment of influenza infection, or that known IFNL immunosuppressive effects can be linked up easily to influenza infection. In stark contrast, there is universal agreement in all publications that IFNab receptors are more ubiquitous, in particular on immune cells, and that IFNa has strong immunostimulatory effects. Differential immunostimulation by IFNa vs. IFNL is therefore the most likely explanation of our findings, and we provide evidence for a variety of immunostimulatory effects of IFNa that IFNL does not have throughout this study.

The authors need to evaluate the role of endogenous type I and type III IFN on IFN therapy. Prior IFN therapy, virus infected mice should be grouped according to their constitutive expression of type I and type III.

Our reply: We agree that measuring endogenous IFN levels is helpful to understand the cytokine environment into which treatment is given. This is now shown in fig. 10A below. As our mice are inbred, the endogenous IFN levels in infected mice on day 2 are very tight and would not allow to group mice into high or low responders at the onset of treatment. In addition, our experimental design includes the prior assignment of mice to treatment group and cohousing of mice undergoing different treatment regimens, to avoid any subjective bias or cage-dependent differences that could impact this study. This experimental design renders a possible grouping after the beginning of the experiment difficult. Last but not least, IFN levels are measured in BAL and lung under terminal anaesthesia, and therefore subsequent grouping of mice for treatment is not compatible with this measurement. We have attempted without success to detect IFNs in the blood, and the results are shown below (Fig. 10B). Hence, grouping of live mice on the basis of blood IFN levels for further treatment is not feasible.

Fig.10: IFN α , IFN β and IFN λ levels in BALF and blood of infected mice at the moment of IFN treatment.

A: BAL fluid

*Fig.10: IFN levels were measured ex vivo in BAL (A) or blood (B) at day 2 of infection, prior to IFN treatment. *, $P < 0.05$; **, $P < 0.01$*

B: Blood

Induction of type I and type III IFN by influenza A virus has different impact on the immune response and related effects on IFN therapy and control of infection.

Our reply: We agree that this is most likely the case. One example is a study by Lasfar et al. (Immunology 2014 (142), 442-52) where interactions between IFN α and IFN γ were assessed, and another the study by Ank et al. (J.Immunol. 2008 (180), 2474-2485) showing that IFNAR1 deficiency reduces IFN transcription upon Sendai virus infection, while IL28Ra deficiency has no effect. Also, Duong et al. (J. Exp. Med. 2014 (211), 857-68) show that hepatocyte upregulate IL28Ra upon IFN α treatment. All these studies were already or are now quoted in our paper (p16, 328-332). As the variable in our study is the IFN type used for treatment, not the mice infected, the endogenous IFN levels are a constant across all groups, and, as shown in Fig. 10, do not spread widely between mice for the three IFNs tested. We have also looked at IFN levels over time post treatment and show the results in Fig. 11 and 12 below. We find increased IFN α protein levels after IFN α treatment, likewise increased IFN λ protein after IFN λ treatment (Fig. 11), but we find no evidence of a positive feedback loop for endogenous IFN production, as transcript levels for all IFNs tested are essentially unaffected by IFN treatments (Fig. 12). We have also looked at IFNAR1 and IL28Ra mRNA levels in our transcriptional profiling of the whole lung, and we find no evidence of an impact of IFN α or IFN λ on the expression levels of these receptors. This is in line with the data from the ImmGen browser shown in Fig. 2A,B in this letter which demonstrate that IFNAR1 and IL28Ra expression do not change significantly by IFN α treatment in most immune cells. Hence, while multiple interactions take place here, we think we have controlled as best as presently possible for accurate distinction between effects of IFN α vs IFN λ treatment.

Fig.11: IFN α , IFN β and IFN λ protein levels in infected mice over time
Fig.11: IFNs were measured in BAL fluid at indicated time points in infected mice treated or not with IFNs. IFN α treatment leads to increased IFN α levels, and IFN λ treatment increases IFN λ levels. *, $P < 0.05$; **, $P < 0.01$, where * indicates IFN α :Veh Ctrl, ° indicates IFN λ :Veh Ctrl, and + indicates IFN α :IFN λ

Fig.12: IFN α , IFN β and IFN λ mRNA levels in infected mice over time
Fig.12: IFN mRNA was measured in total lung RNA at indicated time points in infected mice treated or not with IFNs. Data is normalized to infected, PBS-treated mice at the same time point. Neither IFN treatment leads to statistically significant increases in IFN transcription, suggesting that the increases seen in Fig.11 are due to the detection of exogenous IFN administered and not due to enhancing endogenous IFN production.

Interpretation of the data based on some effects of exogenous IFN is misleading. The mode of administration of IFN is crucial in IFN dosage and related side effects. Prior studies on IFN-alpha intranasal therapy by Tovey group showed a beneficial effect in virus clearance without any significant exacerbation.

Our reply: As the reviewer can see in figures 2 and 4, the novel figure EV2, a great part of the study has been done in infected mice, where endogenous and exogenous IFNs are present. Together with figures 3-12 in this letter, we feel we have abundantly covered the *in vivo* effects of IFN therapy during infection, in the presence of endogenous IFNs. We thank the reviewer for pointing out the Tovey study. We assume that the referee refers to the Tovey et al. paper in JICR 1999: In that study, all IFN treatments are done 1h post infection, which is probably closer to our pre-infection treatment than the post-infection setting where we see deleterious effects by IFN α treatment starting 2 days post infection. We think that our treatment regimen, at day 2 when clinical signs set in, is clinically more relevant than a 1 hour post infection treatment. So we don't see any contradiction between the Tovey paper and our data, since IFN α is not deleterious when infection is aborted early. It is pathogenic only in the context of ongoing IAV infection. The Tovey study is now quoted and discussed in the manuscript on page 13, lines 251-254.

Other concerns: Title is misleading: not really reflecting the data and the role of IFN-lambda.

Our reply: With due respect, we think that the title precisely reflects our findings, which is an absence of pro-inflammatory effects of IFN λ . As discussed extensively above, we have no evidence for active immunosuppressive effects, but we have ample evidence for the absence of increases in inflammatory parameters when treating with IFN λ as compared to treatment with IFN α . We therefore stand by our title and are happy to get advice from the editor whether this title reflects or

not the data we present.

Inconsistent references: The authors need more updates on the advance clinical applications of IFN-lambda.

Our reply: We thank the reviewer for this comment and have now extended the discussion and citations of clinical trials using IFNL on page 16, 320-328.

In conclusion, thanks to the issues raised by the referees we think that the revised manuscript is much improved and clarifies our arguments. We hope to have addressed satisfactorily all concerns and hope that the referees now find the manuscript acceptable for publication in EMBO Molecular Medicine.

2nd Editorial Decision

28 June 2016

Thank you for the submission of your revised manuscript to EMBO Molecular Medicine. We have now received the enclosed reports from the referees that were asked to re-assess it. As you will see the reviewers are now globally supportive and I am pleased to inform you that we will be able to accept your manuscript pending the following final amendments:

Please make sure to discuss referee 1 points about novelty in the paper discussion section, including the concern about the mouse model. Please also provide a point-by-point response to this referee along with the revised article.

Please submit your revised manuscript within two weeks. I look forward to seeing a revised form of your manuscript as soon as possible.

***** Reviewer's comments *****

Referee #1 (Comments on Novelty/Model System):

The results might only apply to this particular mouse model, and therefore the studies are only incremental of what was already known.

Referee #1 (Remarks):

I continue to disagree with the arguments of the authors with respect to novelty. The results might only apply to this particular mouse model, and therefore the studies are only incremental of what was already known.

Referee #2 (Comments on Novelty/Model System):

The authors have shown very strong preliminary data supporting the therapeutic use of IFN-lambda in influenza infection. The technical quality of the work shown is high and the data shown are quite comprehensive and convincing.

While another reviewer questioned the novelty of this report since the results might be expected based on our current understanding of how IFN-lambda differs from IFN-alpha, this is the first demonstration I've seen of IFN-lambda's therapeutic potential in established influenza infection.

The medical impact of this work is clear: IFN-lambda has the potential to be a useful intervention in humans with severe influenza. While IFN-alpha may induce a cytokine storm, and thereby increase mortality, IFN-lambda appears to be well tolerated in human trials.

Another critique of the previous submission was that the authors should validate their findings in a ferret model. The authors respond that such work will require production of ferret IFN-lambda and is really a separate study. I concur with this response and feel that the work presented, incorporating

mice and human tissues, stands on its own.

Another critique of the previous submission related to possible immune-inhibitory effects of IFN-lambda. The authors have included data to address this critique and I find it compelling. In short, the data suggest that the immune cell subsets examined do not show any effects of IFN-lambda.

Referee #2 (Remarks):

I believe that the authors have addressed all of the critiques. This is a very nice study showing efficacy in a mouse model, with extensive data to support the authors' model of how it works. The data are backed up with additional studies in relevant human cell types. The paper is well written and the data stand on their own. The authors do not need the ferret model to make their case, and at any rate ferret experiments will take a long time due to lack of specific ferret reagents.

Referee #3 (Comments on Novelty/Model System):

Well done.

Referee #3 (Remarks):

Thank you for taking into consideration our concerns and improving the manuscript.

2nd Revision - authors' response

08 July 2016

Referee #1 (Remarks):

I continue to disagree with the arguments of the authors with respect to novelty. The results might only apply to this particular mouse model, and therefore the studies are only incremental of what was already known.

Our reply: We have now included a phrase taking into account the referee's concern and the need to confirm this result in additional animal models (lines 264-268).

Corresponding Author Name: Andreas Wack

Journal Submitted to: Embo Molecular Medicine

Manuscript Number: EMM-2016-06413-T